# Identification of Novel Sphydrofuran-Derived Derivatives with Lipid-Lowering Activity from the Active Crude Extracts of *Nocardiopsis* sp. ZHD001

**DOI:** 10.3390/ijms24032822

**Published:** 2023-02-01

**Authors:** Yuhong Tian, Yongjun Jiang, Zhengshun Wen, Liping Guan, Xiaokun Ouyang, Wanjing Ding, Zhongjun Ma

**Affiliations:** 1School of Food and Pharmacy, Zhejiang Ocean University, Zhoushan 316022, China; 2Institute of Marine Biology and Pharmacology, Ocean College, Zhejiang University, Zhoushan 316021, China

**Keywords:** *Nocardiopsis* sp., sphydrofuran derivatives, structure elucidation, lipid-lowering

## Abstract

Lipid-lowering is one of the most effective methods of prevention and treatment for cardiovascular diseases. However, most clinical lipid-lowering drugs have adverse effects and cannot achieve the desired efficacy in some complex hyperlipidemia patients, so it is of great significance to develop safe and effective novel lipid-lowering drugs. In the course of our project aimed at discovering the chemical novelty and bioactive natural products of marine-derived actinomycetes, we found that the organic crude extracts (OCEs) of *Nocardiopsis* sp. ZHD001 exhibited strong in vivo efficacies in reducing weight gain, lowering LDL-C, TC, and TG levels, and improving HDL-C levels in high-fat-diet-fed mice models. Chemical investigations of the active OCEs led to identifying two new sphydrofuran-derived compounds (**1**–**2**) and one known 2-methyl-4-(1-glycerol)-furan (**3**). Their structures were elucidated by the analysis of HRESIMS, 1D and 2D NMR spectroscopic data, and ECD calculations. Among these compounds, compound **1** represents a novel rearranged sphydrofuran-derived derivative. Bioactivity evaluations of these pure compounds showed that all the compounds exhibited significant lipid-lowering activity with lower cytotoxicity in vitro compared to simvastatin. Our results demonstrate that sphydrofuran-derived derivatives might be promising candidates for lipid-lowering drugs.

## 1. Introduction

Cardiovascular diseases (CAs) are the leading cause of premature mortality and disability worldwide, and their incidence and prevalence are likely to increase steadily in the future [1]. It was estimated that 17.9 million people died from CA diseases in 2019, accounting for 32% of all global deaths. Of these deaths, 85% were due to heart attack and stroke [2]. Hyperlipidemia refers to high blood lipid levels, mainly manifested as excessive low-density lipoprotein cholesterol (LDL-C), total cholesterol (TC), and triglyceride (TG), as well as decreased high-density lipoprotein cholesterol (HDL-C) [3]. Clinical studies have shown that hyperlipidemia is a significant independent risk factor for CA, and reducing blood lipid levels can effectively reduce the incidence of CA and significantly improve the quality of life for CA patients [4]. As a consequence, the control of blood lipid levels has become a key objective in the effort to prevent CA events. In recent years, many lipid-lowering drugs have appeared, such as statins, fibrates, bile acid sequestrants, ezetimibe, n-3 fatty acids, and PCSK9 inhibitors. However, most of these lipid-lowering drugs have adverse effects and cannot achieve the desired efficacy in some complex hyperlipidemia patients [5], so it is of great significance to develop safe and effective novel lipid-lowering agents.

Actinomycetes are Gram-positive prokaryotic organisms with high GC DNA content and have proven to be one of the most important natural sources of diverse chemical structures and biologic properties [6,7]. Approximately 70% of the pharmaceutically active secondary metabolites which are currently being used in clinics were discovered from actinomycetes [8,9], including a series of anticancer, antitumor, antibiotics, and immunosuppressive agents. The ocean is a huge body of saltwater that covers nearly 70% of the Earth’s surface, and the marine environment includes tremendous biodiversity [10]. As marine ecosystems, such as seawater, seafloor sediments, and sea mud, are extremely different from terrestrial ones, marine-derived actinomycetes are reported to be a particularly rich source of new chemical structures [11,12,13]. For example, salinosporamide A from marine actinomycete strain *Salinospora* CNB-392 is a potent 20S proteasome inhibitor which was granted an orphan drug designation by the FDA for the treatment of multiple myeloma [14]. Similarly, abyssomicin C, from marine actinomycete strain *Verrucosispora* AB 18-032, is a potent inhibitor of *p*-aminobenzonic acid and inhibits folic acid biosynthesis at an earlier stage than well-known synthetic sulfa drugs [15].

Sphydrofuran is a structurally unique natural product isolated from the culture filtrate of the strain *Streptomyces* sp. MC41-M1 and MC340-A1 by a chemical screening method using Ehrlich’s reagent in 1971 [16,17]. However, the absolute configuration of sphydrofuran was only elucidated by X-ray crystallography and the Helmchen method in 1991 [18] and further confirmed by total synthesis in 1997 [19]. Sphydrofuran is an anomeric and ring–chain tautomeric mixture and could be transformed into stable furan derivative 2-methyl-4-(1-glycerol)-furan and 2*H*-pyran compound under acidic conditions [18,20]. Only three sphydrofuran-derived analogues have been reported so far, some of which have been reported to possess antibacterial and antiviral activities [20,21]. To the best of our knowledge, the lipid-lowering activities of sphdrofuran’s derivatives have never been reported.

During the course of our project aimed at obtaining chemical novelty and bioactive natural products from marine-derived actinomycetes [22,23,24,25], a strain classified as *Nocardiopsis* sp. ZHD001 attracted our attention because its OCE showed strong lipid-lowering activities in vivo. A chemical investigation of the active OCE of *Nocardiopsis* sp. ZHD001 led to the characterizations of two new spydrofuran-derived derivatives, designated as Nocarfurans D-E (**1**–**2**, Figure 1), together with a known analogue, 2-methyl-4-(1-glycerol)-furan (**3**) [18]. Among them, compound **1** represents a novel rearranged sphydrofuran-derived derivative, which might be a rearrangement from known compound **3** by ring reformation, dehydration, reduction, and acetylation. Furthermore, bioactivity evaluations of these pure compounds showed that all the compounds exhibited lipid-lowering activity with lower cytotoxicity in vitro compared to simvastatin. In this paper, we describe the isolation and culture of strain *Nocardiopsis* sp. ZHD001, the preparation and bioactive evaluation of OCE, and the isolation, structure identification, proposed biosynthetic pathway, and bioactive evaluation of these isolated compounds.

## 2. Results and Discussion

### 2.1. Preparation of OCE and Its Hypolipidemic Effects In Vivo

The pure colony of *Nocardiopsis* sp. ZHD001 (Appendix A) growing on Gause’s agar plate was inoculated into a 500 mL Erlenmeyer flask, containing 250 mL of Gause’s liquid medium, and then incubated at 28 °C for four days on a rotary shaker (180 rpm) to create seed broth. The 4 mL of seed broth was then transferred to into a 500 mL Erlenmeyer flask, containing 250 mL of Gause’s liquid medium, and then cultured for ten days at the same conditions for the culture of seed broth. The whole fermented cultures (100 L) were extracted with ethyl acetate at room temperature three times. The resulting ethyl acetate part was dried in a vacuum to obtain the OCE (10.6 g) for this study.

To evaluate the hypolipidemic efficacy of OCE in vivo, the C57BL/6 mice fed with a high-fat diet (HFD) and normal diet (ND) were used as a hyperlipemia model and control group, respectively. After 12 weeks of feeding, the body weights of the mice in the HFD group were significantly higher than those in the ND group. In comparison to the mice in the HFD group, the mice from the intragastric administration of OCE (HFD-OCE) group lost weight during the administration (Appendix A). In addition, the effects of OCE on serum biochemical indexes, including LDL-C, TC, TG, and HDL-C, are shown in Figure 2. The results showed that serum LDL-C, TC, and TG levels in the mice of the HFD group were significantly higher than those in the ND group, although the HDL-C significantly increased. In comparison to the HFD group, OCE significantly reduced the serum LDL-C levels by 61.4%, TC levels by 10.6%, and TG levels by 65.4%, as well as increasing HDL-C levels by 55.7%. Above all, these results indicate that OCE exhibits hypolipidemic effects in vivo.

### 2.2. Isolation and Structure Elucidation of Compounds ***1***–***3***

In order to identify the active ingredients, the remaining active OCEs (5.4 g) were separated by repeated silica gel and Sephadex LH-20 column chromatography and further purified by high-performance liquid chromatography (HPLC) to afford two new compounds **1**–**2**, as well as the known compound **3** (Figure 1).

Compound **1** was isolated as a yellowish amorphous solid from methanol. The HRESIMS data of **1** exhibited an [M + Na]^+^ ion peak at *m*/*z* 237.0741 (Appendix A), suggesting a molecular formula of C_10_H_14_O_5_, in accordance with four degrees of unsaturation. The strong absorption peaks of the IR spectrum (Appendix A) revealed the existence of C=O (1736 cm^−1^) and -OH (3396 cm^−1^). Thorough analysis of ^1^H NMR data (Table 1 and Appendix A) of **1** revealed one characteristic trisubstituted furan ring at δ_H_ 6.01 (1H, s), two methyl groups [δ_H_ 2.24 (3H, s), δ_H_ 2.02 (3H, s)], two methylenes [δ_H_ 3.71 (1H, dd, *J* = 11.4, 5.4 Hz) and 3.78 (1H, dd, *J* = 10.8, 7.2 Hz), δ_H_ 4.98 (2H, s)], and one oxygenated methine [δ_H_ 4.76 (1H, dd, *J* = 7.2, 6.0 Hz)]. The ^13^C NMR data (Table 1 and Appendix A) of **1** revealed ten carbon signals including one trisubstituted furan ring [δ_C_ 108.7, 119.7, 151.6 and 152.7] and one acetyl group (δ_C_ 20.8 and 172.8). Compound **1** was assigned as the derivative of the 1D NMR combined with previous studies of co-isolated known compound 2-methyl-4-(1-glyceryl)-furan (**3**) [18].

The planar structure of **1** was deduced by combined interpretation of ^1^H-^1^H COSY, HSQC, and HMBC correlations (Figure 3 and Appendix A). A 2,4,5-trisubstitued furan ring was established by the HMBC correlations of H-3 to C-2, C-5, and C-4, respectively. The HMBC correlations of H-1 to C-2, C-3, C-5, and C-4 suggested that the methyl C-1 was connected to C-2. The HMBC correlations of H-8 to C-2, C-3, C-4, C-5, and C-9, and H-10 to C-9 indicate the fragment -CH_2_OOCCH_3_ was attached at C-4. Finally, based on the HMBC correlations of H-6 to C-7, and H-7 to C-6 and C-5, as well as the COSY correlations of H-6 to H-7, combined with the missing hydrogen and oxygen in the molecular formula, the -CHOHCH_2_OH fragment attached to C-5 was deduced. Thus, the planar structure of compound **1** was elucidated, as shown in Figure 3.

The absolute configuration of **1** was determined by computational methods. Firstly, conformational analyses of **1** were carried out via Spartan 10 software using molecular mechanisms with an MMFF force field with a Boltzmann population of over 5%. Subsequently, the selected conformers of **1** were reoptimized at the B3LYP/6-31^+^g (d, p) level in methanol by the Gaussian 09 program. The ECD spectra were simulated by the overlapping Gaussian function (δ = 0.35 eV), in which velocity rotatory strengths of the first 60 exited states were calculated. In order to obtain the final ECD spectra, the simulated spectra of the lowest energy conformers were averaged according to the Boltzmann distribution theory and their relative Gibbs free energy (ΔG). The results showed that the calculated CD curves of (6*R*)-**1** showed good agreement with the experimental ones (Figure 4). Thus, the absolute structure of **1** was determined as shown and named Nicardifuran D.

Compound **2** was obtained as a yellowish oil with the molecular formula of C_10_H_14_O_5_ based on the HREISMS data at *m*/*z* 237.0741 [M + Na]^+^ (Appendix A), corresponding to four degrees of unsaturation. The IR spectrum of **2** (Appendix A) showed the existence of C=O (1735 cm^−1^) and -OH (3418 cm^−1^).The ^1^H NMR (Table 1 and Appendix A) of **2** confirmed two methyl groups [δ_H_ 2.24 (3H, s), δ_H_ 2.04 (3H, s)], one methylene [δ_H_ 3.92 (1H, dd, *J* = 10.8, 6.6 Hz) and 4.10 (1H, dd, *J* = 11.4, 3.6 Hz)], and four methines [δ_H_ 6.05 (1H, s), 7.31 (1H, s), 4.51 (1H, d, *J* = 6.0 Hz), 3.86 (1H, m)]. The ^13^C NMR (Table 1 and Appendix A) analysis displayed 10 carbon signals in total at δ_C_ 13.4, 20.7, 68.6, 66.9, 73.8, 106.1, 127.8, 139.6, 153.9, and 172.9. A comparison of the above 1D NMR data of 2 with those of 2-methyl-4-(1-glyceryl)-furan (**3**) revealed that the structures were similar, except for the presence of an acetyl group in **2** at C-8, which was consistent with the molecular formula and further confirmed by key ^1^H-^1^H COSY, HSQC and HMBC correlations (Figure 3 and Appendix A). Both 2 and co-purified 2-methyl-4-(1-glyceryl)-furan (**3**) had positive specific rotation and shared similar biosynthetic origins; supporting the absolute configuration of 2 were 6*R* and 7*R*, in accordance with **3**. Thus, the structure of **2** was determined as shown and named as Nicardifuran E.

Compound **3** was obtained as a colourless amorphous solid with the molecular formula C_8_H_12_O_4_ as deduced by HRESIMS (*m*/*z* 195.0628 ([M+Na]^+^ and Appendix A). The 1D NMR data analyses (Table 1, Appendix A), combined with a specific rotation value, led to the identification of **3** as 2-methyl-4-(1-glyceryl)-furan, which was previously reported from a *Streptomyces* sp. MC41-M1 and MC340-A1 [16].

### 2.3. Proposed Biosynthetic Pathways of Compounds ***1***–***3***

Biosynthetically, compounds **1**–**3** may be traced back to sphydrofuran, which is an anomeric and ring–chain tautomeric mixture. Sphydrofuran could easily be transformed into the stable furan derivative **3**, starting with an elimination of H_2_O and followed by ring-opening elimination. Compound **1** was thought to be generated from **3**, and the process most likely involves dehydration, ring reformation, and acetylation. The direct acetylation of compound **3** at C-8 may lead to the formation of **2** (Figure 5).

### 2.4. Effect of Compounds ***1***–***3*** on Lipid Accumulation in Fatty-Acid-Elicited HepG2 Cells

In order to investigate the effect of compounds **1**–**3** on the reduction in lipid accumulation in vitro, we utilized the lipid-loaded model of HepG2 cells exposed to 0.45 mM fatty acid (oleic acid:palmitic acid = 2:1) for 24 h. Firstly, we determined that compounds **1**–**3** did not exhibit significant toxicity to HepG2 cells at a high concentration of 160 μM. Next, we demonstrated that compounds **1**–**3** can inhibit the lipid droplet accumulation caused by fatty acid treatment in HepG2 cells at the concentration of 10 μM (Figure 6 and Appendix A). Quantitative analysis exhibited that compounds **1**–**3** at the concentrations of 10 μM reduced the intracellular fat deposition by 7.4%, 17.1%, and 10.1%, respectively (Figure 7A). The effects of compounds **1**–**3** were similar to that induced by the positive control simvastatin (Figure 7A). In addition, intracellular LDL-C, TG, and TC contents were detected in HepG2 cells. As shown in Figure 7B–D, LDL-C, TG, and TC levels were significantly increased due to fatty acid treatment. Compounds **1**–**3** at the concentration of 10 μM had significantly decreased intracellular LDL-C levels by 4.7%, 18.7%, and 58.9% (Figure 7B), TG levels by 54.7%, 64.2%, and 63.4% (Figure 7C), TC levels by 6.5%, 1.0%, and 33.0%, respectively (Figure 7D). These results indicate that compounds **1**–**3** can effectively ameliorate lipid accumulation in fatty-acid-elicited HepG2 cells.

## 3. Materials and Methods

### 3.1. General Experimental Procedures

C57BL/6J male mice (SPF) at 6–8 weeks were obtained from Beijing Vital River La-boratory Animal Technology Co., Ltd., Beijing, China. All mice were housed in the specific pathogen-free facility at the School of Food and Pharmacy of Zhejiang Ocean University in compliance with the guidelines and regulations of the Experimental Animal Care and Use Committee of Zhejiang Ocean University. The mice were placed in separate mouse cagesin a room maintained under 12/12 h light−dark cycles with free access to water and food. In addition, the temperature of the housing environment was controlled at 22–24 °C and relative humidity 40~70%. Mice were allowed to adapt to their new environment for 7 days to habituate steady metabolic conditions. Then, the ND group mice were fed with normal diet and distilled water intragastric administration. The HFD group mice were fed with high-fat diet and distilled water intragastric administration. The HFD-OCE group mice were fed with high-fat diet and the OCE intragastric administration (10 mg/kg/d). The high-fat diet was Research Diets D12492 (60 kcal% Fat, Suzhou Shuangshi Experimental Animal Feed Technology Co., Ltd., Suzhou, China). Mice were monitored for body weight and food intake for 12 weeks. After that, the mice were fasted overnight and blood samples were collected for serum preparation and stored at −20 °C. After blood sampling, all the mice were euthanized. The LDL-C, TC, TG, and HDL-C lipid levels in the serum were tested using a commercial assay kit (Nanjing Jiancheng Bioengineering Institute, Nanjing, China).

Silica gel (100–200 mesh or 300–400 mesh, Qingdao Ocean Chemical Company, Qingdao, China) and Sephadex LH-20 (Amersham Pharmacia Biotech, Japan) were used for column chromatography and further purified by preparative HPLC (SHIMAZU LC-20AP) equipped with an Agilent Pursuit C-18 column (10 μm, 21.2 × 250 mm). Components were detected by TLC under UV light (254 or 365 nm), and spots were visualized by heating silica-gel plates sprayed with 8–10% H_2_SO_4_ in ethyl alcohol. Then, 1D and 2D NMR data were recorded in Methanol-*d*_4_ (δ_H_ 3.31, δ_C_ 49.0) using a JEOL NMR spectrometer (600 MHz for ^1^H NMR and 150 MHz for ^13^C NMR; JEOL, Tokyo, Japan). Chemical shifts were expressed in δ (ppm) and coupling constants (*J*) in Hz. Optical rotations were measured on an AUTOPOL I digital polarimeter (Rudolph Research Analytical, Hackettstown, NJ, USA). UV and ECD spectra were measured on a Shimadzu UV-1800 and a JASCO J-1500-150ST spectrophotometer, respectively. IR spectra were acquired on a Thermo Scientific Nicolet iS10 FT-IR spectrometer. HRESIMS data were obtained on a 6230 Time of Flight Liquid Chromatography/Mass Spectrometry (TOF LC/MS) spectrometer (Agilent Technologies, Palo Alto, CA, USA).

The HepG2 cells were obtained from the Cell Bank of the Chinese Academy of Sciences, Shanghai, China. Foetal bovine serum (FBS), Minimum Essential Medium (MEM), Phosphate-buffered saline (PBS), Trypsin–EDTA (0.25%), Penicillin–streptomycin, Alanine acid, *L*-glutamic acid, and MEM Non-Essential Amino Acids were acquired from Gibco (Thermo Scientific). Sodium oleate and Sodium palmitate, Simvastatin, and Oil Red O were obtained from Sigma (St. Louis, MO, USA). A 4% PFA Fix Solution was purchased from Beyotime Biotechnology Institute, Shanghai, China. The TG assay kit, TC assay kit, and LDL-C assay kit were purchased from Nanjing Jiancheng Bioengineering Institute.

### 3.2. Isolation and Identification of Strain Nocardiopsis sp. ZHD001

The titled strain (Appendix A) was isolated from marine sediments acquired from Zhoushan Island by using a standard dilution plating method. The title strain was identified using a 16S rDNA sequence analysis by TaKaRa Biotechnology (Dalian, China) Co., Ltd., and its DNA sequence was compared to the GenBank database using BLAST (nucleotide sequence comparison). A BLAST search result exhibited the sequence of the strain to have 99.68% similarity with *Nocardiopsis* sp. AD16 (accession number MF410670.1), indicating that the strain belonging to the genus *Nocardiopsis* sp. The sequence of the strain is deposited in GenBank under accession no. MN736491.

### 3.3. Isolation and Purification of Compounds ***1***–***3***

The remaining active OCEs (5.4 g) were packed into a silica gel column (50 cm × 6.0 cm) eluting with CH_2_Cl_2_–MeOH (100:1, 50:1, 30:1, 20:1, 10:1, 5:1, 1:1, and 0:1, each 1 L) to produce five fractions (A–E). Fraction D (175 mg) was purified by preparative HPLC (17–60% MeOH in H_2_O, 40 min, flow rate at 10 mL/min) to produce **3** (21.7 mg, t_R_ = 17.9 min) and fraction D1 (13.5 mg). Fraction D1 was then further separated by preparative HPLC (20–100% MeOH in H_2_O, 40 min, flow rate at 10 mL/min) to produce **1** (1.8 mg, t_R_ = 25.0 min). Fraction E (580 mg) was purified by preparative HPLC (20–100% MeOH in H_2_O, 40 min, flow rate at 10 mL/min) to produce **2** (6.7 mg, t_R_ = 19.0 min).

Nicardifuran D (**1**): yellowish powder (MeOH); [*α*]^20^_D_ +12.0 (*c* 0.1, MeOH); ^1^H and ^13^C NMR data, see Table 1; HRESIMS *m*/*z* 237.0741 [M + Na]^+^ (calcd for C_10_H_14_NaO_5_, 237.0733).

Nicardifuran E (**2**): yellowish powder (MeOH); [*α*]^20^_D_ +40 (*c* 0.1, MeOH); ^1^H and ^13^C NMR data, see Table 1; HRESIMS *m*/*z* 237.0733 [M + Na]^+^ (calcd for C_10_H_14_NaO_5_, 237.0733).

2-methyl-4-(1-glyceryl)-furan (**3**): yellowish oil (MeOH); ^1^H and ^13^C NMR data, see Table 1; HRESIMS *m*/*z* 195.0632 [M + Na]^+^ (calcd for C_8_H_12_NaO_4_, 195.0628).

### 3.4. Cytotoxicity Assay and Lipid-Lowering Activity Assay In Vitro

HepG2 cells were maintained in Minimum Essential Medium containing 10% FBS, 1% Penicillin–streptomycin, 1% Alanine acid, and 1% L-glutamic acid at 37 °C in a 5% CO_2_ incubator. The cells were digested with Trypsin–EDTA (0.25%) every 3 days, cell passage was carried out at 1:2, and cells at the logarithmic stage were selected for the experiment.

Cytotoxicity was tested by an MTT assay. HepG2 was inoculated in 96-well culture plates with about 10,000 cells per well at 37 °C in a 5% CO_2_ incubator for 24 h. The control group and compound **1**–**3** groups with the final concentration were set, with 4 parallel wells in each group. The culture was continued for 24 h. Then, 20 uL of a 0.5 mg/mL MTT solution was added to each well and cultured for a further 4 h before the medium was removed and 150 uL of DMSO was added to each well. We then placed the 96-well culture plates on the cell oscillator for 10 min and obtained the optical density (OD) at 490 nm with an enzyme marker. Cell survival rate = OD **1**–**3** group 490 nm/OD control group 490 nm × 100%.

Lipid droplets were observed in an Oil Red O experiment. HepG2 was inoculated in 12-well culture plates with about 300,000 cells per well at 37 °C in a 5% CO_2_ incubator for 24 h. The control group, model group, positive control group, and compound **1**–**3** groups were set, with 4 parallel wells in each group. Then, 1mL of blank MEM was added to the control group, A total of 1 mL of sodium oleate (0.6 mM) and sodium palmitate (0.3 mM) mixture to the model group, and 0.5 mL of the mould agent mixture and 0.5 mL of the compound corresponding concentration solution to the compound administration group. After 24 h, each well was rinsed 3 times with PBS and fixed with 4% PFA Fix Solution for 40 min per well. Each well was washed with PBS 3 times and stained with Oil Red O working solution for 40 min. Each well was washed with PBS 2 times. The staining of lipid droplets in cells was observed under a microscope.

## 4. Conclusions

Marine-derived actinomycetes are important sources for the discovery of novel bioactive natural products. In this paper, we concluded that the organic crude extracts of a marine actinomycete, *Nocardiopsis* sp. ZHD001, can significantly reduce the levels of plasma TC, TG, and LDL-C and increase the plasma HDL-C level in high-fat-diet-fed mice. Then, we isolated two new sphydrofuran-derived derivatives (**1**–**2**) and 2-methyl-4-(1-glycerol)-furan (**3**) from the active organic crude extracts. Among these compounds, compound **1** represents a novel rearranged sphydrofuran-derived derivative. In addition, all of these compounds were screened for their cytotoxic and lipid-lowering activities in vitro. The results showed that compounds **1**–**3** did not exhibit significant toxicity to HepG2 cells at a high concentration of 160 μM. In addition, compounds **1**–**3** can inhibit the accumulation of lipid droplets effectively in HepG2 cells at a concentration of 10 μM, with compound **2** showing itself to be especially superior to simvastatin. In addition, compounds **1**–**3** had significantly reduced the intracellular LDL-C, TG, and TC levels at a concentration of 10 μM. Above all, these findings suggest that the discoveries of these sphydrofuran-derived derivatives with lower cytotoxicity and lipid-lowering activities could be of great importance to the development of new lipid-lowering agents.

## Figures and Tables

**Figure 1 ijms-24-02822-f001:**
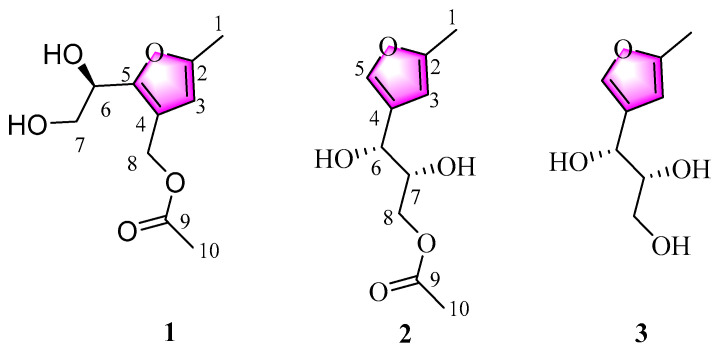
Chemical structures of compounds **1**–**3**.

**Figure 2 ijms-24-02822-f002:**
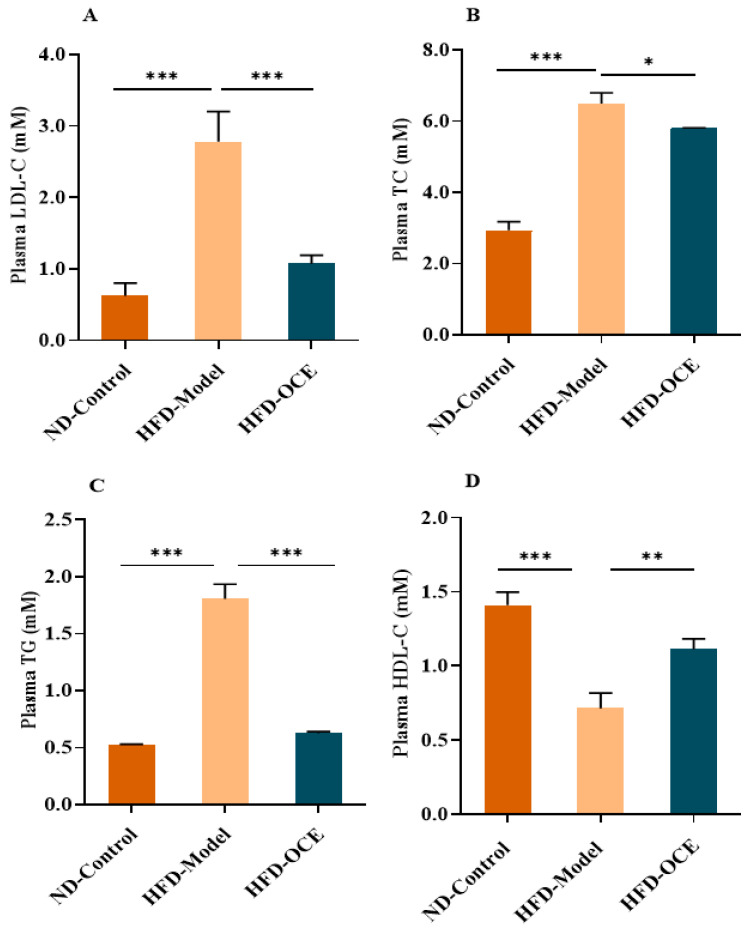
Effects of OCE on the serum lipid levels in mice. (**A**) LDL-C concentration (mM, *n* = 4). (**B**) TC concentration (mM, *n* = 4). (**C**) TG concentration (mM, *n* = 4). (**D**) HDL-C concentration (mM, *n* = 4). Mice were fed with normal diet (ND) or high-fat diet (HFD) with or without OCE at the dose of 10 mg/kg by gavage once a day for 12 weeks. Values were the mean ± standard deviation (SD), * *p* < 0.05, ** *p* < 0.01, *** *p* < 0.001 (one-way ANOVA).

**Figure 3 ijms-24-02822-f003:**
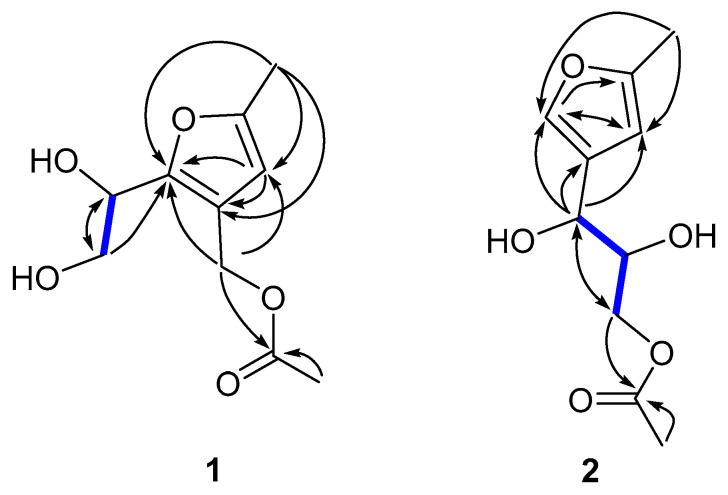
Key COSY (bold lines) and HMBC (arrows) correlations of compounds **1**–**2**.

**Figure 4 ijms-24-02822-f004:**
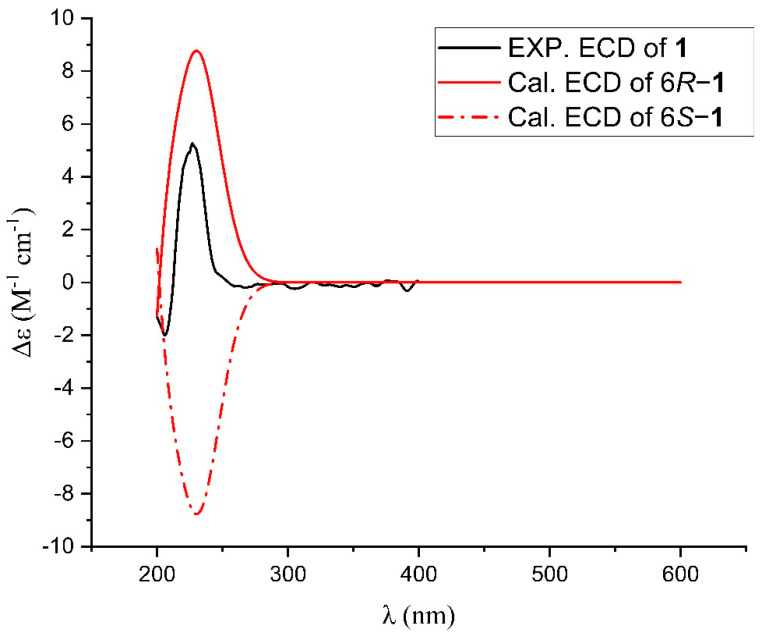
Experimental and calculated ECD spectra for **1**.

**Figure 5 ijms-24-02822-f005:**
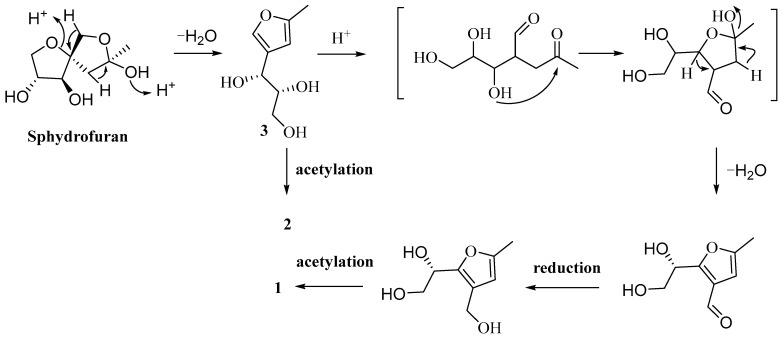
Proposed biosynthetic pathways for compounds **1**–**3**.

**Figure 6 ijms-24-02822-f006:**
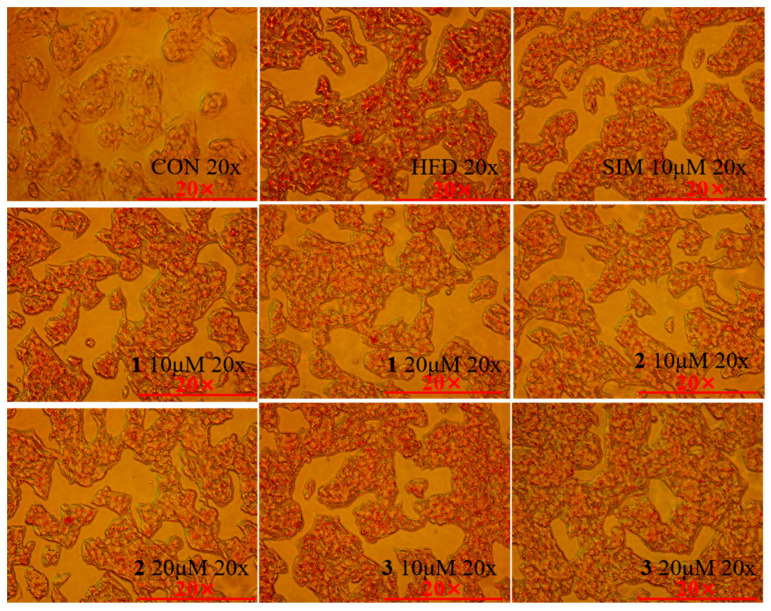
Oil red O of HepG2 cells under 20× mirror. CON: cells without any treatment. HFD: cells treated with high fatty acids alone. SIM 10 μM: cells treated with high fatty acids and simvastatin 10 μM. Sets of samples: cells treated with high fatty acids and corresponding sample concentrations.

**Figure 7 ijms-24-02822-f007:**
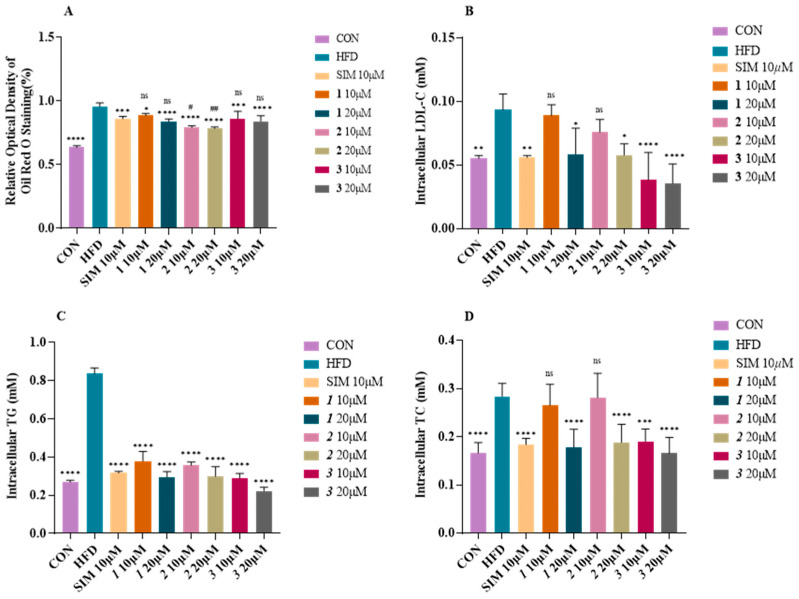
(**A**) Quantification analysis by Oil Red O staining. Data were the mean ± standard deviation (SD) by Prism; each group was compared with HFD group; * *p* < 0.05, ** *p* < 0.01, *** *p* < 0.001, and **** *p* < 0.0001 (one-way ANOVA). Every group except group CON and group HFD was compared with SIM 10 μM group, ^ns^
*p* > 0.05, ^#^
*p* < 0.05, ^##^
*p* < 0.01 (one-way ANOVA). Intracellular LDL-C (**B**), TC (**C**), and TG (**D**). Data were the mean ± standard deviation (SD) by Prism; each group was compared with HFD group; ^ns^
*p* > 0.05, * *p* < 0.05, ** *p* < 0.01, *** *p* < 0.001, and **** *p* < 0.0001 (one-way ANOVA).

**Table 1 ijms-24-02822-t001:** ^1^H (600 MHz) and ^13^C (150 MHz) NMR data for compounds **1**–**3** recorded in Methanol-*d*_4_.

No.	1		2		3	
	δ_H_, *J* in Hz	δ_C_, type	δ_H_, *J* in Hz	δ_C_, type	δ_H_, *J* in Hz	δ_C_, type
1	2.24, s	13.3, CH_3_	2.24, s	13.4, CH_3_	2.24, s	13.4, CH_3_
2		152.7, C		153.9, C		153.7, C
3	6.01, s	108.7, CH	6.05, s	106.1, CH	6.06, s	106.1, CH
4		119.7, C		127.8, C		128.2, C
5		151.6, C	7.31, s	139.6, CH	7.30, s	139.4, CH
6	4.76, dd (7.2, 6.0)	67.9, CH	4.51, d (6.0)	68.6, CH	4.52, d (6.0)	68.3, CH
7	3,71, dd (11.4, 5.4) 3.78, dd (10.8, 7.2)	65.6, CH_2_	3.86, m	73.8, CH	3.42, dd (10.8, 6.0)	76.4, CH

8	4.98, s	58.6, CH_2_	3.92, dd (10.8, 6.6) 4.10, dd (11.4, 3.6)	66.9, CH_2_	3.56, dd (10.8, 4.2) 3.67, m	64.1, CH_2_
9		172.8, C		172.9, C		
10	2.02, s	20.8, CH_3_	2.04, s	20.7, CH_3_		

## Data Availability

Not applicable.

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
