# Peer review of "Identification of Novel Sphydrofuran-Derived Derivatives with Lipid-Lowering Activity from the Active Crude Extracts of Nocardiopsis sp. ZHD001"

_ijms, 2023, doi:10.3390/ijms24032822_

Round 1

Reviewer 1 Report

A manuscript entitled “Identification of novel sphydrofuran-derived derivatives with lipid-lowering activity from the active crude extracts of Nocardipsis sp. ZHD001” was submitted by Tian et al for publication in the International Journal of Molecular Sciences. First the authors describe the extraction and structure determination of three potentially active molecules. In the last part of the manuscript the bioactivity of these compounds is described. This fragment of the manuscript is too short and not informative. The authors show 2 figures (figure 6 and 7) which are described in three sentences. This is not sufficient and shall be improved before the manuscript may be accepted for publication. A minor review is required.

Author Response

For reviewer ’s comments:

A manuscript entitled “Identification of novel sphydrofuran-derived derivatives with lipid-lowering activity from the active crude extracts of Nocardipsis sp. ZHD001” was submitted by Tian et al for publication in the International Journal of Molecular Sciences. First the authors describe the extraction and structure determination of three potentially active molecules. In the last part of the manuscript the bioactivity of these compounds is described. This fragment of the manuscript is too short and not informative. The authors show 2 figures (figure 6 and 7) which are described in three sentences. This is not sufficient and shall be improved before the manuscript may be accepted for publication. A minor review is required.

  • Thank you for your valuable suggestion, we have carefully revised the manuscript based on your comments and the bioactivity of  compounds' description had been revised in the following 

“In order to investigate the effect of compounds 1-3 on reduction of lipid accumulation in vitro, we utilized the lipid-loaded model of HepG2 cells exposed to 0.5 mM fatty acid (oleic acid: palmitic acid = 2: 1) for 24 hours. First, we determined that compounds 1-3 did not exhibit significant toxicity to HepG2 cells at a high concentration of 160 μМ. Next, we demonstrated that compound 1-3 can inhibit the lipid droplets accumulation caused by fatty acid treatment in HepG2 cells at the concentration of 10 μM (Figure 6). Quantitative analysis exhibited that compounds 1-3 at the concentrations of 10 μM reduced the intracellular fat deposition by 7.4%, 17.1% and 10.1%, respectively (Figure 7A). The effect of compounds 1-3 were similar to that induced by the positive control simvastatin (Figure 7A). Besides, intracellular LDL-C, TG and TC contents were detected in HepG2 cells. As shown in Figure 7B-7D, LDL-C, TG and TC levels were significantly increased due to fatty acid treatment. Compounds 1-3 at the concentration of 10 μM had significantly decreased intracellular LDL-C levels by 4.7%, 18.7%, 58.9% (Figure 7B), TG levels by 54.7%, 64.2%, 63.4% (Figure 7C), TC levels by 6.5%, 1.0%, 33.0%, respectively (Figure 7D).”

Reviewer 2 Report

The manuscript numbered 2153745 deals with the searching the novel hypolipidemic substances of marine origin, its purification, identification of structure, putative biosynthesis pathways and characterization of its biological properties.

This paper may be of interest for the journal’ audience. The paper focus on the recent scientific matters and also provide some practical aspects to fighting with the cardiovascular diseases. Manuscript is well written, the need of the research is enough justified, experiment was well thought out, planned and executed. Materials and methods are exhaustively described, used methods are novel and well-chosen to achieve the main aim of the study. Obtained results were sufficiently described discussed, discussion are explanatory and informative. Concise conclusion sum up the most important achievements of this paper.

Author Response

Thank you for your valuable suggestion, We have carefully revised the manuscript based on your comments.